# Evaluation of Yield Improvements in Machine vs. Visual Strength Grading for Softwood Species

Gonzalo Moltini [1,2,*], Guillermo Íñiguez-González [1], Gonzalo Cabrera [2] and Vanesa Baño [2,3]

1 Escuela Técnica Superior de Ingeniería de Montes, Forestal y del Medio Natural,
Universidad Politécnica de Madrid, 28040 Madrid, Spain
2 CESEFOR, 42005 Soria, Spain
3 Faculty of Engineering, Universidad de la República, Montevideo 1130, Uruguay
* Correspondence: g.moltini@gmail.com

**Abstract:** The current national standard for strength grading in Spain is based on a visual classification, which, for softwood species with small cross-sections (with a thickness equa tol or less than 70 mm), establishes two different visual grades (ME-1 and ME-2). These grades are assigned to the strength classes C24 and C18, respectively, for maritime and radiata pines, and C27 and C18 for Scots pine, according to the European standard EN-1912:2012. The production of engineered wood products, such as glulam or cross-laminated timber is increasing worldwide. The machine grading of wood using non-destructive testing provides the industry with a more reliable, fast, and consistent method for grading. With this background in mind, this study presents the yield comparison of machine grading vs. visual grading for those three pine species from Spain. The machine settings were obtained according to the standard EN 14081-2:2019, providing several possible strength grade combinations. Results allow new possibilities for the industry and improve the structural yield of the studied timber, thus increasing the material optimization.

**Keywords:** timber; pines; machine strength grading; visual strength grading; structural yield

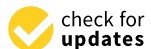



## 1. Introduction

The use of timber as structural material requires the assignment of mechanical properties, and this can be done either by visual or mechanical grading according to national and European standards.

Visual grading consists of the measurement of visual singularities, such as knots, fissures, grain deviation, etc., and relating them with the values of strength, modulus of elasticity, and density of the wood species, which are obtained from experimental tests. Density can be also a parameter used for visual grading. National standards currently in use for visual grading in Spain are UNE 56544:2022 [1] and UNE 56546:2022 [2] for softwood and hardwood species, respectively. These standards are discussed in subcommittee 6 of structural timber of the technical committee of wood and cork of AENOR (AENOR CTN 56 SC-6). Currently, three visual grades (ME-1, ME-2, and MEG) can be assigned to four pine species with Spanish provenance: maritime pine (*Pinus pinaster* Ait.), radiata pine (*Pinus radiata* D. Don), Scots pine (*Pinus sylvestris* L.), and Salzsmann pine (*Pinus nigra* ssp. *Salzsmannii*); and one visual grade (MEF) for two hardwood species: sweet chestnut (*Castanea sativa* Mill.) and Southern blue gum (*Eucalyptus globulus* Labill.), named according to EN 13556:2003 [3]. The correspondence between the visual grades and the strength classes of EN 338:2016 [4] is given by EN 1912:2012 [5].

Machine grading provides a better and faster classification method, thus, using manual or visual grading is not reliable in terms of productivity [6,7]. It consists of measuring one or more Indicating Properties (IPs), such as the dynamic elastic modulus and/or density among others, by using non-destructive techniques. The equipment to be used

in the machine grading must be approved by the corresponding European Normalization Committee to obtain the settings to be implemented in the machines used by the industry for structural grading. As in the case of visual grading, Ips are correlated with the physical and mechanical properties from bending or tensile mechanical tests of sawn timber specimens of structural size. The standard that regulates the mechanical grading equipment and derivation of settings is EN 14081-2:2019 [8] The settings for each species and origin are confidentially discussed and approved by the Technical Group TG1 from CEN/TC124/WG2 (Standardization European Committee-Technical Committee 124 of timber structures-Working Group 2 of Solid Timber). Those settings are later given to the manufacturers of each machine grading equipment through AGR documents (Approved Grading Report), which provide different combinations of strength classes per species and origin and the corresponding settings to be implemented in the homologated machine grading equipment. From 2021, as a result of the work carried out by Cesefor in collaboration with Brookhuis (Enschede, Netherlands) and Microtec (Bressanone, Italy) [9–14], three softwood species (maritime pine, radiata pine, and Scots pine) and one hardwood species (Southern blue gum) from Spain are included in AGR documents and available for machine grading in the industry. Settings were derived for both handheld (MTG 960 and MTG 920 from Brookhuis; and Viscan Portable from Microtec) and in-line grading machines (mtgBATCH 922, mtgBATCH 926, mtgBATCH 962, mtgBATCH 966 from Brookhuis; and Viscan, Viscan Compact, Goldeneye 702 and Goldeneye 706 from Microtec).

However, even though CE marking depends on the strength classes provided by the homologated grading machines and collected in AGR documents, there are many research works relating non-destructive measurements with the physical and mechanical properties. Llana et al., 2020 [15] present a review of the non-destructive testing techniques used in Spain, providing statistical linear models for estimating the mechanical and physical properties of sawn timber

The traditional use for these species in Spain is associated with heavy timber framing, where large cross-sections are used to conform to a post and beam structural system. However, in recent years a great interest in the production of Engineered Wood Products (EWPs) was observed in this country, with several industries manufacturing Glued Laminated Timber (GLT) and Cross-Laminated Timber (CLT) from national softwood species (maritime pine, radiata pine, and Scots pine). GLT and CLT are manufactured by gluing sawn boards or lamellas with thicknesses up to 45 mm, Figure 1. Large-scale production of EWPs cannot depend on visual grading or portable machine grading due to problems associated with grading speed and costs associated with personnel.

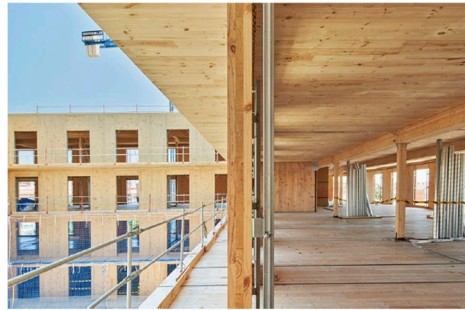 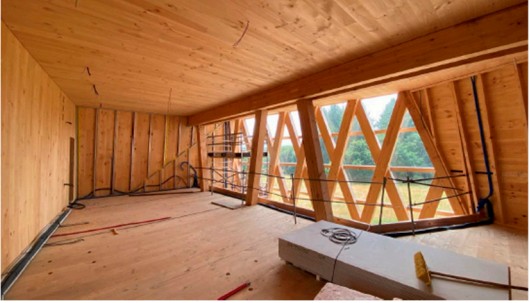

**Figure 1.** Residential building in Cornellá and Green Impulse Building in Lugo, build using GLT and CLT of Radiata pine (Source: Ref. [16]).

Since machine strength grading provides faster and better results than visual grading, the objective of the present study is to quantify the yield improvement in the assignation of strength classes for maritime pine, radiata pine, and Scots pine from Spain, the distribution of these species can be seen in Figure 2.

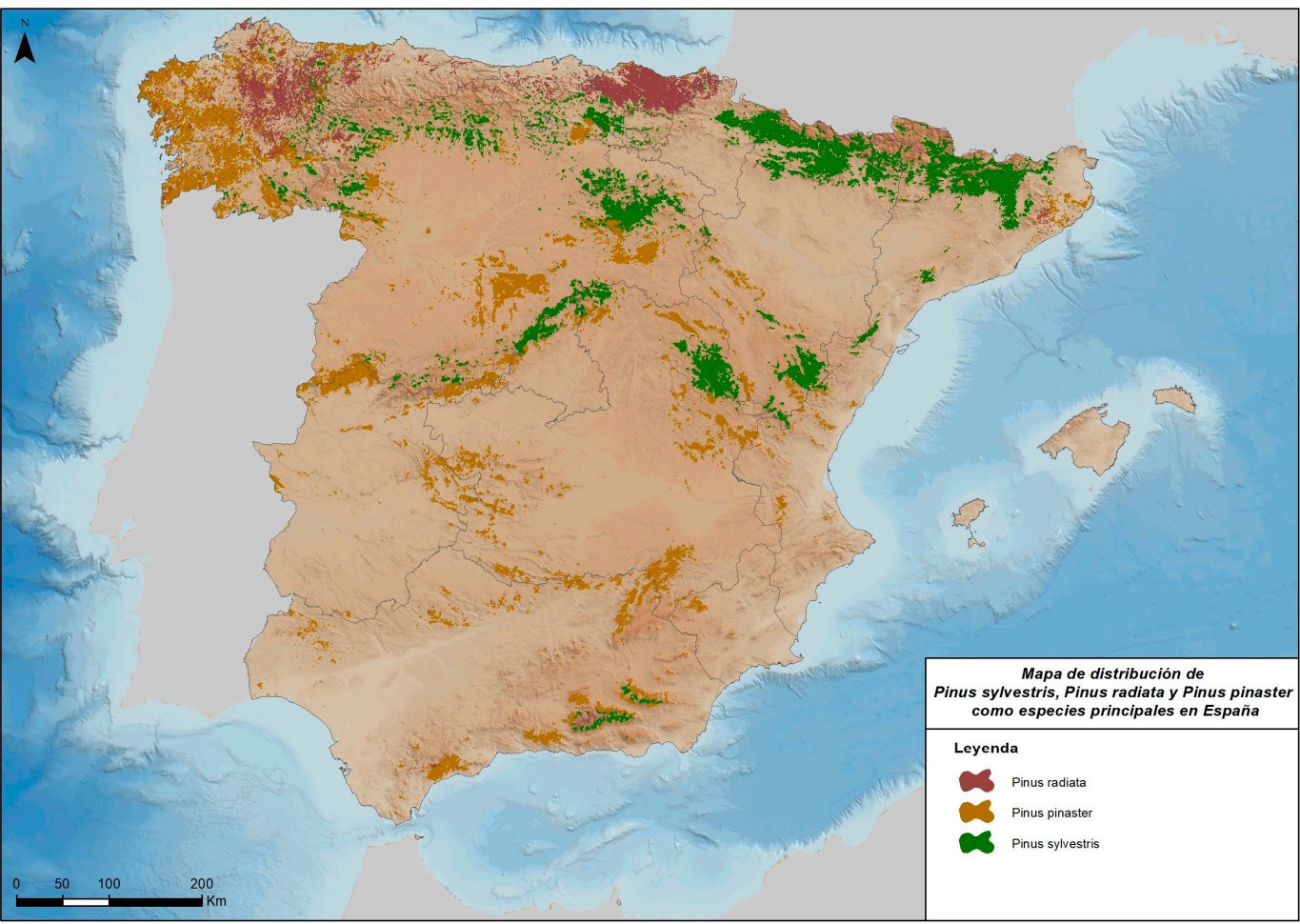

**Figure 2.** Distribution of maritime, radiata, and Scots pine in Spain [17].

## 2. Methodology

### 2.1. Sampling

The sampling for each species includes four different areas, each subsample with more than 100 pieces and the total sample size for each species is greater than 450 specimens, as required by EN 14081-2:2019 [8]. For the definition of the origins, the four areas with greater wood volume output were selected for each species [17] to represent the timber to be graded in the production line. Radiata and Scots pine specimens were sampled from sawmills, where specimens were selected randomly based on the commonly commercialized stock. As for maritime pine, specimens were obtained from a forest sampling (7 plots from origins A, B, and C and 3 plots from origin D). All the final specimens used for grading and testing fulfilled the requirements of EN 14081-1:2016 [18]. The origin and sample size for each subsample are shown in Table 1 and Figure 3.

There are two identified subspecies of maritime pine in Spain, *P. pinaster* ssp. *atlantica* and ssp. *mesogensis*, with different mechanical and physical properties. They are in the northern regions, matching the Atlantic climate, and in the interior and Mediterranean regions, with a continental type of climate, respectively [19,20]. Since the production of EWPs relies on the spp. *atlantica*, this subspecies was the one selected for the present study.

**Table 1.** Subsamples origin and number of specimens (size).

| Species | Subsample A | | Subsample B | | Subsample C | | Subsample D | | Total |
|---|---|---|---|---|---|---|---|---|---|
| | Origin | N | Origin | N | Origin | N | Origin | N | N |
| Maritime pine (ssp. *atlantica*) | Galicia North and Asturias | 122 | Galicia West | 121 | Galicia Interior | 129 | Basque Country | 111 | 483 |
| Radiata pine | Asturias | 109 | Galicia | 116 | Gipuzkoa-Basque Country | 145 | Biscay-Basque Country | 125 | 495 |
| Scots pine | Segovia-Castile and Leon | 147 | Soria-Castile and Leon | 165 | Cuenca- Castile La Mancha | 119 | Navarre | 128 | 559 |

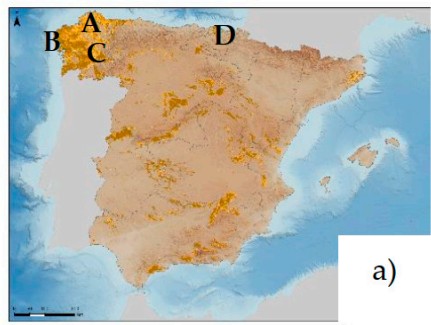 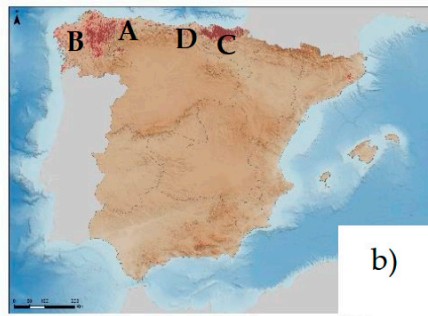 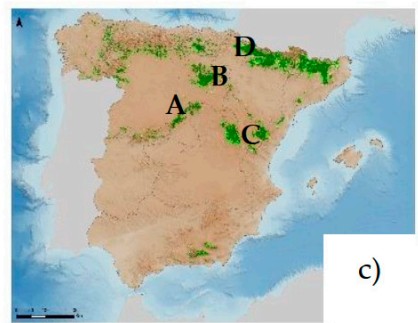

**Figure 3.** Identification of subsamples by species: (**a**) maritime pine; (**b**) radiata pine; and (**c**) Scots pine.

### 2.2. Visual and Mechanical Grading

Specimens were visually graded in visual grades: ME-1 and ME-2, according to the Spanish standard UNE 56544:2022 [1], based on the relative size of the knots (both in width and thickness), maximum growth ring size, length of fissures, warp, wane, rot, and damage by insects. For machine grading, the frequency of the first mode for longitudinal vibration was measured using the handheld machine MTG 960 (Brookhuis) at the Cesefor facilities (Soria, Spain) as shown in Figure 4a. In addition, dimensions, weight, and moisture content were measured. These data were used to derive both the settings for in-line and hand-held equipment as the frequency and weight readings are the same for every equipment.

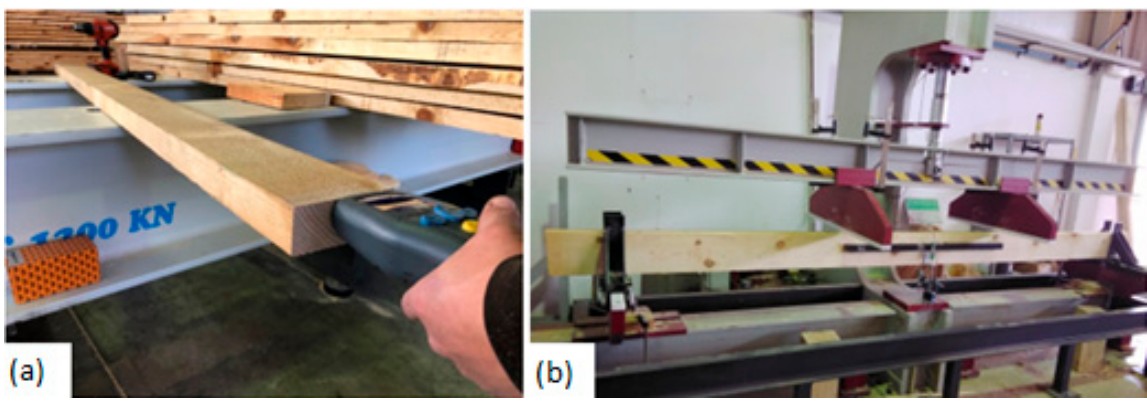

**Figure 4.** Machine grading (**a**) and bending test (**b**).

### 2.3. Mechanical and Physical Properties

After visual and machine grading, four-point bending tests were performed according to EN 408:2010/A1:2012 [21]. The critical section was located within the pure bending span (positioned centrally when possible) according to EN 384:2016+A2:2019 [20]. Figure 4 shows the setup of handheld machine grading using MTG960-Brookhuis (Figure 4a) and bending tests (Figure 4b).

Modulus of elasticity parallel to the grain $(E_{0,m})$ was calculated from the mean value of the local displacements, measured as the mean value obtained by two extensometers located on both sides of the beams in the central third of the beam according to the longitudinal direction; and bending strength $(f_m)$ was calculated from the ultimate failure load.

The moisture content was determined according to the standard EN 13183-1:2002/A:2003 [22] from a slice of the specimen obtained close to the failure point, as well as the density of each specimen $(\rho)$. The modulus of elasticity and the density were later adjusted to a reference moisture content of 12% according to EN 384:2016+A2:2019 [20] and bending strength to a reference width of 150 mm. Finally, the characteristic values of the physical and mechanical properties were calculated according to EN 14358:2016 [23], which were named Grade Determining Properties (GDP).

### 2.4. Indicating Property (IP)

Since dynamic modulus of elasticity was identified as a good predictor for stiffness, bending, and tensile strength in softwood species [24,25], the Indicating Property used for the assignation of a strength class from non-destructive measurements was this parameter, determined as shown in Equation (1).

$$IP = (2\,L\,f_0\,CF)^2\,\rho_{12}\,\times\,10^{-6},\tag{1}$$

where, $L$ is the total length of the specimen; $f_0$ is the frequency for the first longitudinal vibration mode; $CF$ is a correction factor given by Ravenshorst and Van de Kuilen [26], which adjusts the wave speed to the 12% of moisture content as shown in Equation (2); and $\rho_{12}$ is the density considering the weight and dimensions of the whole specimen at a reference moisture content of 12 % (adjusted by EN 384:2016+A2:2019 [20]).

$$CF = \begin{cases} 1 & u \leq 12\% \\ \left(1 - \frac{0.1(u-12)}{13}\right)^{-1} & 12\% < u \leq 25\%, \\ (0.9)^{-1} & u > 25\% \end{cases}\tag{2}$$

where, $u$ is the moisture content of the specimen.

### 2.5. Derivation of Settings

Settings to be implemented in the machine grading per species were derived with the aim to assign a strength class to each value of IP, according to EN 14081-2:2019 [8]. For that, the required characteristic values of strength, modulus of elasticity, and density for the sample were defined as those of each strength class considered, EN 338:2016 [4]), except for the case of modulus of elasticity, in which the required value is calculated as $E_{0,mean,EN338} \times 0.95$, for C strength classes. Then, the characteristic values of GDP were calculated according to EN 384:2016+A2:2019 [20] and EN 14358:2016 [23]. Characteristic values of strength are calculated as $f_k = k_v\,f_{05}$, being $f_{05}$ the 5th percentile of the strength and $k_v$ the factor that considers the lower variability between subsamples in machine grading with respect to visual grading (EN 384:2016+A2:2019, [20]). $k_v$ is taken as 1 in the case of portable machines and in-line grading machine from bending tests when $f_{m,k} > 30\,\text{N/mm}^2$. For the rest of the cases, $k_v$ is taken as 1.12. Characteristic values of modulus of elasticity are calculated as $E_{0,mean} = \overline{E}_0/0.95$, where $\overline{E}_0$ is the average modulus of elasticity of the sample.

An Iterative process was carried out to obtain the settings for each strength class combination, as shown in Figure 5 and as described following:

(1) Assigned grades for the sample. The sample is graded from preliminary *IP* limit values, and the characteristic values are calculated for each assigned strength class and compared with required values, with the condition that a minimum of 20 specimens are allocated in each grade and the number of the rejected specimens is greater than the maximum value between 5 or 0.5% of the total number of specimens.

(2)　Assigned grades for the subsample. The sample is divided into the subsamples defined in Table 1 (A, B, C and D) and graded according to the preliminary IP values defined in (1). The characteristic values per subsample are calculated as described in (1) and compared with the required values, which for the case of subsample are calculated as 90%, 95%, and 90% of the required values of the sample for strength, modulus of elasticity and density, respectively.

(3)　Optimum grading. In parallel to points (1) and (2), the optimum grading is defined. It consists in assigning the best possible strength class to each specimen while optimizing the yield for the highest strength class. For that, the specimens are sorted from lower to higher values of GDPs. The specimens with the lower values are removed until the characteristic values of the remaining subgroup comply with the required values, which, in this case, are defined directly by the characteristic values of the strength classes (EN 338:2016 [4]).

(4)　Size matrix. The size matrix provides the number of specimens in the optimum and assigned grades for the total sample.

(5)　Elementary cost matrix. The elementary cost matrix provides a cost of efficiency and safety for each preliminary IP limit values, i.e., wrongly downgrading a specimen leads to an efficiency cost, and wrongly upgrading a specimen leads to a safety cost.

(6)　Global cost matrix. The global cost matrix is defined to assess the performance of the grading machine, calculated as the multiplication between the values of the size matrix and elementary cost matrices, and divided by the number of specimens in the assigned grade. The values below the diagonal of the matrix must not exceed 0.4 and trying to minimize the values above the diagonal.

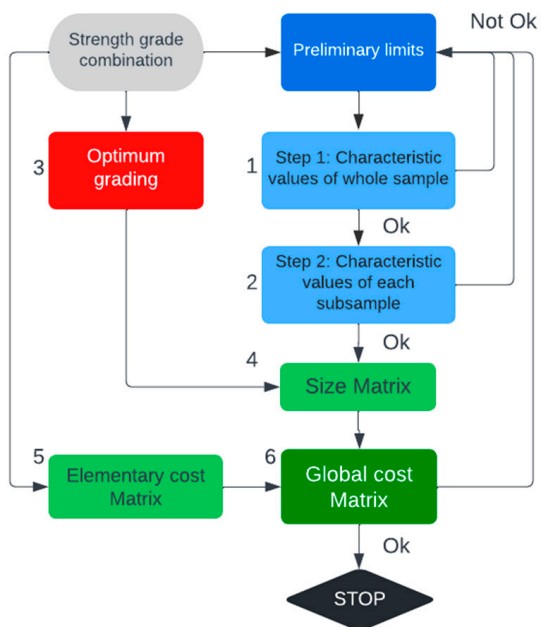

**Figure 5.** IP limits derivation process.

Finally, there is a validation of the IP limits. If the preliminary IP values verify the previous requirements, and the grading yield cannot be increased, are taken as final IP limit values. If not, new preliminary values should be considered (point 1) and the process repeated.

It should be noticed that for all the cases, the criterion to define the strength class combinations was to increase the percentage of pieces assigned to the higher strength class, instead of decreasing the percentage of rejection.

## 3. Results and Discussion

### 3.1. Physical and Mechanical Properties of the Whole Sample

Results of the characteristic values of bending strength, modulus of elasticity, and density obtained from experimental bending tests made according to EN 408:2010/A1:2012 [21] for the whole sample, complying with the requirement of visual features provided by EN 14081-1:2016 [18], are shown in Tables 2–4 for maritime pine, radiata pine, and Scots pine, respectively.

**Table 2.** Characteristic values of the mechanical and physical properties of maritime pine.

| Property | | | Subsample | | | | Total |
|---|---|---|---|---|---|---|---|
| | | | **A** | **B** | **C** | **D** | |
| | | | **Galicia North & Asturias** | **Galicia West** | **Galicia Interior** | **Basque Country (BC)** | |
| Specimens | | | 122 | 121 | 129 | 111 | 483 |
| Moisture content | | % | 11.9 | 12.2 | 12.4 | 12.0 | 12.1 |
| CoV | | % | 6 | 17 | 16 | 26 | 17 |
| Strength | $f_m$ | N/mm$^2$ | 48.7 | 42.0 | 38.3 | 47.0 | 43.9 |
| | $f_{m,k}$ | N/mm$^2$ | 19.0 | 16.1 | 16.2 | 17.7 | 16.9 |
| | CoV | % | 37 | 46 | 45 | 40 | 17 |
| Modulus of elasticity | $E_{0,mean}$ | kN/mm$^2$ | 12.1 | 10.8 | 11.7 | 13.3 | 12.0 |
| | CoV | % | 30 | 38 | 32 | 29 | 33 |
| Density | $\rho_{mean}$ | kg/m$^3$ | 575 | 540 | 540 | 569 | 556 |
| | $\rho_k$ | kg/m$^3$ | 489 | 447 | 446 | 482 | 457 |
| | CoV | % | 9 | 13 | 13 | 11 | 12 |

**Table 3.** Characteristic values of the mechanical and physical properties of radiata pine.

| Property | | | Subsample | | | | Total |
|---|---|---|---|---|---|---|---|
| | | | **A** | **B** | **C** | **D** | |
| | | | **Asturias** | **Galicia** | **Gipuzkoa-BC** | **Biscay-BC** | |
| Specimens | | | 109 | 116 | 145 | 125 | 495 |
| Moisture content | | % | 14.9 | 13.6 | 10.6 | 12.2 | 12.7 |
| CoV | | % | 8 | 10 | 16 | 23 | 20 |
| Strength | $f_m$ | N/mm$^2$ | 28.7 | 47.6 | 36.3 | 38.4 | 37.8 |
| | $f_{m,k}$ | N/mm$^2$ | 12.1 | 22.1 | 16.7 | 18.5 | 15.2 |
| | CoV | % | 44 | 25 | 36 | 35 | 38 |
| Modulus of elasticity | $E_{0,mean}$ | kN/mm$^2$ | 7.65 | 10.2 | 12.1 | 12.4 | 10.8 |
| | CoV | % | 38 | 23 | 22 | 21 | 30 |
| Density | $\rho_{mean}$ | kg/m$^3$ | 479 | 545 | 488 | 501 | 503 |
| | $\rho_k$ | kg/m$^3$ | 417 | 459 | 401 | 409 | 413 |
| | CoV | % | 9 | 11 | 11 | 12 | 12 |

The whole sample before grading shows values equivalent to a strength class C16, C14, and C18 for maritime pine, radiata pine, and Scots pine, respectively. In an analysis of variance (ANOVA), it was observed that there were significant differences ($p < 0.05$ and $F > F_{crit}$) between the three species for the three GDPs evaluated. In an analysis of variance (ANOVA) between subsamples for each species, it was found that there were significant differences between subsamples for the three species and for the three GDPs.

**Table 4.** Characteristic values of the mechanical and physical properties of Scots pine.

| Property | | | Subsample | | | | Total |
|---|---|---|---|---|---|---|---|
| | | | A | B | C | D | |
| | | | Segovia–Castile Leon | Soria–Castile Leon | Cuenca–Castile La Mancha | Navarra | |
| Specimens | | | 147 | 165 | 119 | 128 | 559 |
| Moisture content | | % | 14.3 | 13.6 | 13.6 | 9.1 | 12.8 |
| CoV | | % | 10 | 15 | 15 | 8 | 20 |
| Strength | $f_m$ | N/mm$^2$ | 35.1 | 37.1 | 39.8 | 45.4 | 39.1 |
| | $f_{m,k}$ | N/mm$^2$ | 17.6 | 19.4 | 17.7 | 19.2 | 18.4 |
| | CoV | % | 32 | 28 | 37 | 41 | 20 |
| Modulus of elasticity | $E_{0,mean}$ | kN/mm$^2$ | 10.8 | 11.5 | 10.7 | 12.2 | 11.3 |
| | CoV | % | 19 | 20 | 22 | 23 | 22 |
| Density | $\rho_{mean}$ | kg/m$^3$ | 526 | 531 | 482 | 595 | 534 |
| | $\rho_k$ | kg/m$^3$ | 450 | 464 | 397 | 517 | 441 |
| | CoV | % | 10 | 8 | 11 | 9 | 12 |

### 3.2. Visual Grading

The three samples were visually graded according to UNE 56544:2022 [1] and the characteristic values of bending strength, modulus of elasticity, and density for each visual grade (ME-1, ME-2, and Rejection) were obtained according to a non-parametric determination following the standard EN 14358:2016 [23], as stated by EN 384:2016/A2:2019 [20]. Table 5 shows the characteristic values for the three grades of the whole sample of maritime pine and Scots pine and of the subsamples from Asturias and Galicia for radiata pine.

**Table 5.** Characteristic values of the mechanical and physical properties for each visual grade.

| Species | GDPs | | Visual Grades | | |
|---|---|---|---|---|---|
| | | | ME-1 | ME-2 | R |
| Maritime pine | n | | 72 | 270 | 141 |
| | $f_{m,k}$ | N/mm$^2$ | 25.3 | 16.7 | 14.8 |
| | CoV | % | 32 | 41 | 51 |
| | $E_{0,mean}$ | kN/mm$^2$ | 14.4 | 11.7 | 10.9 |
| | CoV | % | 30 | 34 | 33 |
| | $\rho_k$ | kg/m$^3$ | 517 | 448 | 446 |
| | CoV | % | 9 | 12 | 11 |
| | Strength class (SC) | | C24 | C16 | C14 |
| | SC EN1912 | | C24 | C18 | R |

**Table 5.** *Cont.*

| Species | GDPs | | Visual Grades | | |
|---|---|---|---|---|---|
| | | | ME-1 | ME-2 | R |
| Radiata pine | n | | 68 | 106 | 51 |
| | $f_{m,k}$ CoV | N/mm$^2$ % | 23.1 23 | 12.3 40 | 9.2 61 |
| | $E_{0,mean}$ CoV | kN/mm$^2$ % | 11.6 24 | 9.74 35 | 7.68 44 |
| | $\rho_k$ CoV | kg/m$^3$ % | 449 11 | 414 12 | 404 13 |
| | Strength class (SC) | | C22 | - | - |
| | SC EN1912 | | C24 | C18 | R |
| Scots pine | n | | 134 | 307 | 118 |
| | $f_{m,k}$ CoV | N/mm$^2$ % | 24.5 33 | 19.5 32 | 14.4 43 |
| | $E_{0,mean}$ CoV | kN/mm$^2$ % | 12.7 22 | 11.0 20 | 10.4 30 |
| | $\rho_k$ CoV | kg/m$^3$ % | 422 13 | 406 14 | 387 14 |
| | Strength class (SC) | | C24 | C18 | C14 |
| | SC EN1912 | | C27 | C18 | R |

Strength classes assigned to the visual grades differ from those provided by EN 1912:2012 [5]. On one hand, specimens of maritime pine and Scots pine visually graded as rejection could be assigned to a strength class C14. On the other hand, the strength classes assigned to the visual grade ME-1 was lower than those provided by EN 1912:2012 [5] (C22 instead of C24 for radiata pine and C24 instead of C27 for Scots pine). Finally, the strength classes assigned to the visual grade ME-2 were lower for maritime pine (C16 instead of C18) and it was not possible to assign a strength class of radiata pine, while in EN 1912:2012 [5] is C18. It should be noticed that the lower size of the radiata pine sample compared with those of the other two species could be affecting the results.

*3.3. Machine Grading*

Characteristic values of the strength class combinations complied at least with the required values described in Section 2.5. Figure 6 presents the yields (%) for the different strength class combinations of maritime pine, radiata pine, and Scots pine obtained for in-line grading machine (mtgBATCH 966). C1, C2, and C3 identify the three strength classes in a combination sorted from high (C1) to low (C3) strength class, while R represents the percentage of rejection in the assignation of the strength classes. In the case that only two strength classes are defined by combination, only C2 and C3 appears; and C3 appears when only one strength class is assigned.

In a comparison of the strength class combinations obtained from machine grading between species, the following results were observed. Firstly, the higher yield in terms of the number of strength classes by combination was obtained for Scots pine (six combinations of 3-strength classes), followed by maritime pine (two combinations of 3-strength classes), and only one for radiata pine. Secondly, the better yield in terms of the higher strength class reached was for Scots pine, with 6% of the sample graded as C35. Thirdly, a lower percentage of rejection was also obtained for Scots pine, with only 1% of rejection in all the strength class combinations. Finally, an analysis of the maximum percentage of rejection identified the strength class combination C24/C18 as that with the higher rejection for

radiata pine (27%) and maritime pine (19%), while the lower percentages of rejection were obtained for the combination of only one strength class.

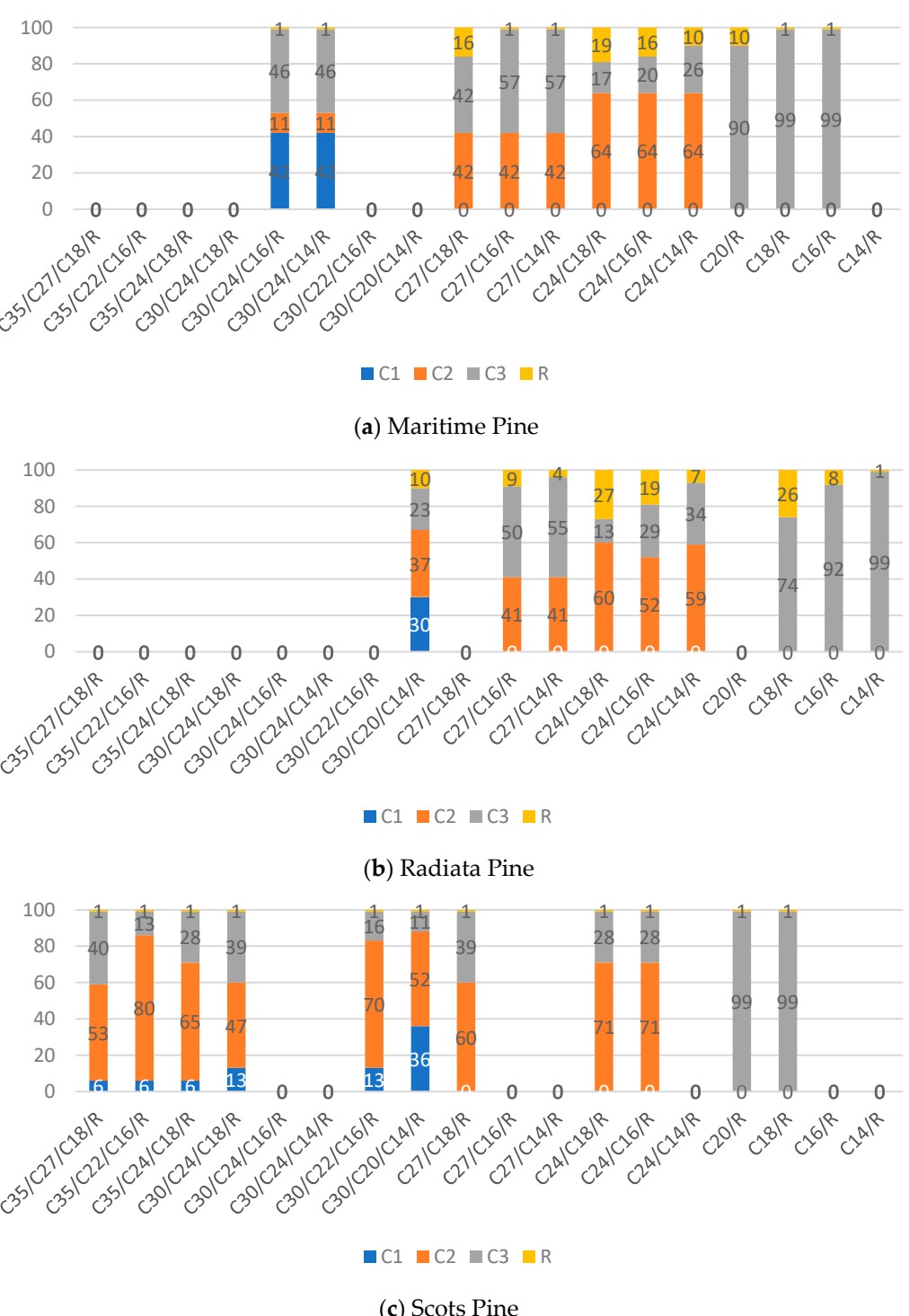

(**a**) Maritime Pine

(**b**) Radiata Pine

(**c**) Scots Pine

**Figure 6.** Yields (%) obtained for in-line grading machines by strength class per each species. Note: Empty columns reflect that the strength class combination did not provide a better yield with respect to the immediately higher strength class combination.

In the same way, Figure 7 presents the yields (%) for the different strength class combinations of maritime, radiata and Scots pine obtained for handheld grading machine (MTG 960).

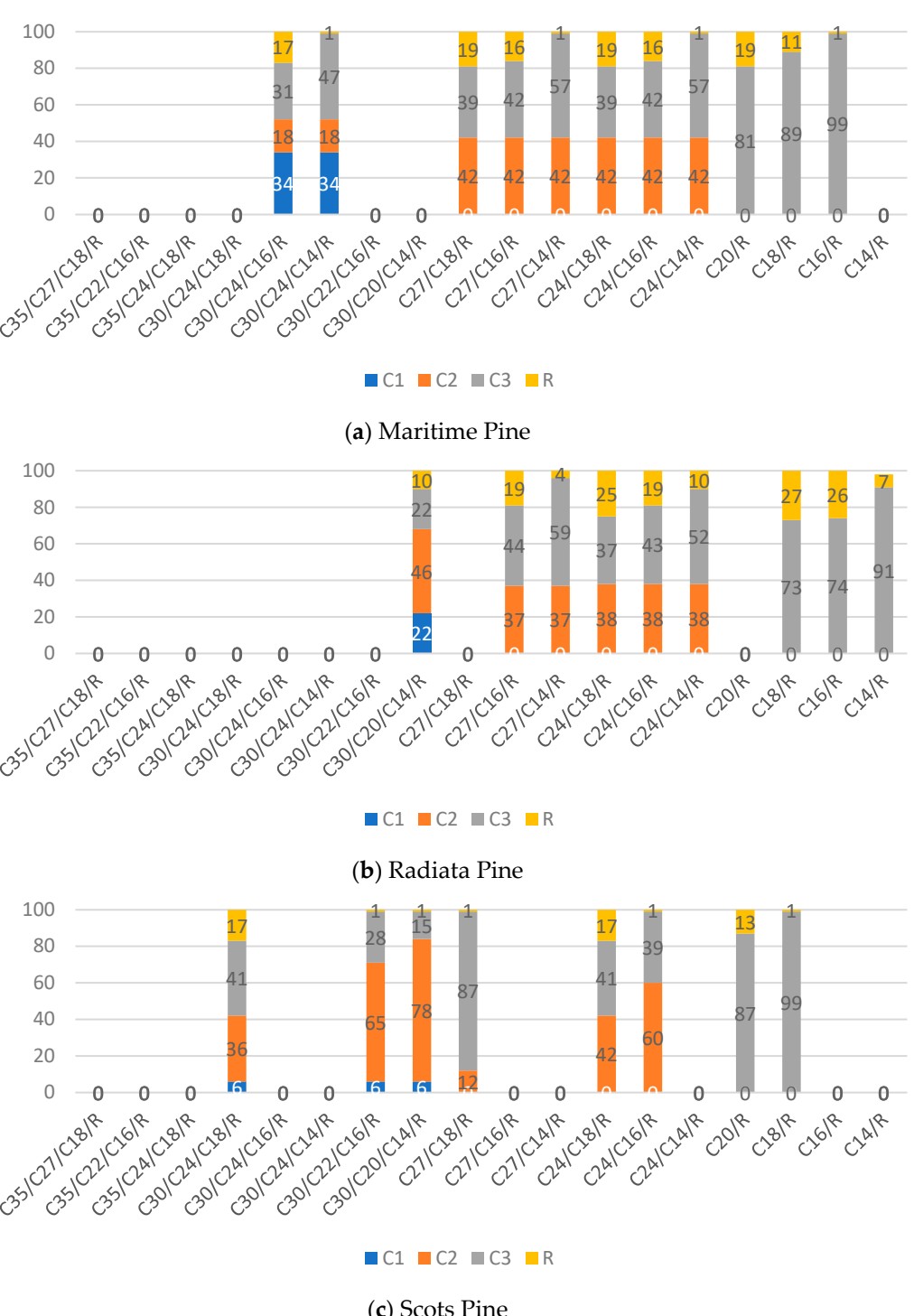

**Figure 7.** Yields (%) obtained for handheld grading machines by strength class by species. Note: Empty columns reflect that the strength class combination did not provide a better yield with respect to the immediately higher strength class combination.

In general, handheld, and in-line machine grading resulted in the same number of strength class combinations and in the same higher strength class reached (C30) for maritime and radiata pines. However, for Scots pine, the number of strength class combinations were lower for handheld than in-line machine grading (8 vs. 11), as well as the higher strength class reached (C35 in-line and C30 in handheld grading).

Regarding the mean value of the percentage of rejections, maritime and radiata pine resulted in 4% higher rejection for handheld than for in-line grading, and Scots pine

resulted in 6% higher. However, the percentage of rejection depends on the strength class combination considered, being the maximum increment of percentage of rejection (16%) reached for Scots pine in the combination C24/C18.

For all the cases, it was observed that combinations with strength classes close to each other resulted in lower yields of the lower strength class and higher percentage of rejections, in agreement with that Olofsson et al. [27] reached for Scots pine in Sweden. For example, for maritime pine, the yield of the strength class combination C27/C16/R resulted in values of 42%/57%/1% for in-line grading while the yield for C24/C16/R resulted in values of 64%/20%/16%; that is, a decrease of 22% of the yield of the second strength class and an increase of 15% of rejected specimens.

### 3.4. Machine Grading vs. Visual Grading

It can be observed that machine grading provides a higher number of strength class combinations than visual grading.

With the aim to quantify the better yield of machine grading with respect to the current visual grades defined in the Spanish standard (UNE 56544:2022 [1]), a comparison between the percentage of specimens assigned to each C strength classes for both visual and machine grading (in-line and handheld) was made. Figure 8 shows the comparison between visual and machine grading for the studied pines for the strength class combination C24/C18/R for maritime and radiata pines and C27/C18/R for Scots pine. It should be noticed that visual grading of radiata pine considered only the subsamples of Asturias and Galicia, and not the whole sample, as the case of the other two species.

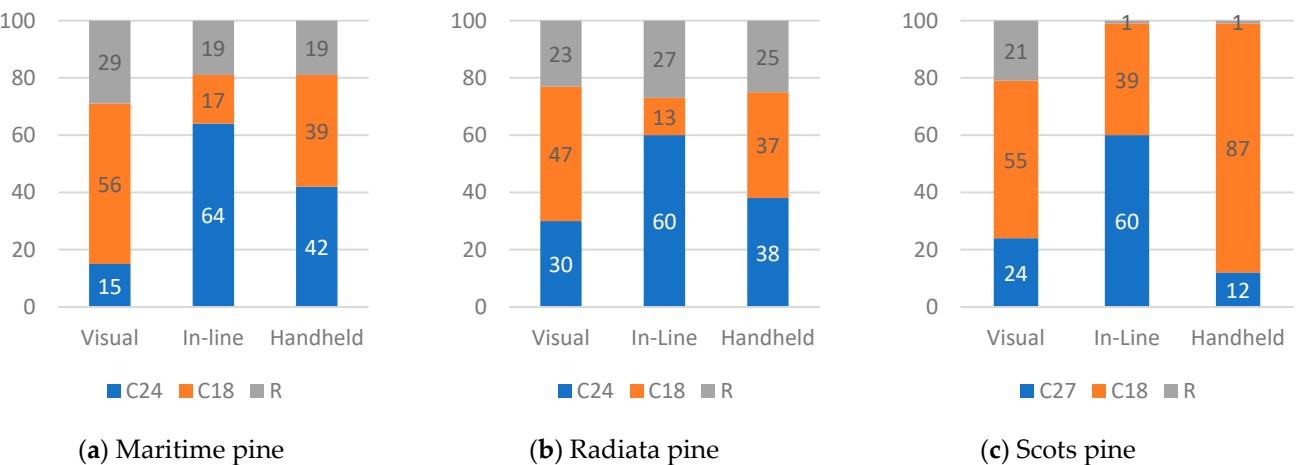

**Figure 8.** Yields (%) comparison between visual and machine grading for C classes.

Even though the visual grading of the studied samples did not comply with the relationships between visual grade and strength class given by EN 1912:2012 [5] for these species, a comparison between machine and visual grading was made considering the theoretical strength classes. As it is observed in Figure 8, for all the species, in-line machine grading showed better yields in the percentage of assignation to the higher strength class than both handheld machine grading and visual grading. However, the handheld grading yield of the higher strength class for Scots pine was lower than that from the visual grading.

As expected, and other authors had concluded [28], in maritime pine and Scots pine, in addition to the better yield of the higher strength class, a decrease in the percentage of rejection was obtained for machine grading with respect to visual grading. Visual grading of maritime pine resulted in similar yields than that obtained by Vega et al. [29] for this species from Asturias, with 6% of the sample graded as C24, 65% as C18, and 29% of rejection, showing that machine grading improved the yield in both percentage of rejection and assignation to the higher strength class.

In radiata pine, even though the yield in the assignation of the higher strength class of machine grading was higher than that of visual grading, the percentage of rejection

hardly changed. However, percentages of rejection in both visual and machine grading of radiata pine were lower than those obtained by Hermoso et al. [30] and Vega et al. [29]. Hermoso et al. [30] declared that 21% of a sample from Basque Country could be assigned to a strength class C24 and 42% to C18, while the rejection rate was 37%. On the other hand, results from Vega et al. [29] were even lower, with 76% radiata pine from Asturias assigned to the strength class C18 and a rejection percentage of 24%, with no pieces assigned to the highest strength class (C24).

Results of visual grading evidenced the limitations of the method, similar conclusion to those obtained by Nocetti et al. [31], stating that knots and density as individual parameters did not show a good correlation with timber strength, making it necessary to involve other parameters, such as dynamic modulus of elasticity, to improve the prediction.

## 4. Conclusions

Significant differences in the physical and mechanical properties between pine species from Spain and between origins by species were obtained. The higher strength class assigned to the whole sample, without previous visual or machine grading, was obtained for Scots pine and the lower for radiata pine.

Visual grading according to the Spanish standard UNE 56544 resulted in strength classes different to those provided by EN 1912, which led to think about the need to review the standard for these species.

In machine grading, it is possible to prioritize the yield of the higher strength class or the percentage of rejection, resulting in different strength class combinations.

In-line and handheld machine grading resulted in the same number of strength class combinations and the same higher strength class assigned for maritime and radiata pine. However, the in-line machine grading resulted in a number of strength class combinations 38% higher than in handheld machine grading for Scots pine, as well as in a higher possible strength class.

It was observed that to define combinations with strength classes close to each other resulted in a lower grading yield, i.e., for the same yield in the higher strength class, there is an increase of the rejections.

In general, the yield of machine grading was higher than that of visual grading in terms of the number of strength class combinations, the higher strength class reached, the percentage of specimens classified in the higher strength class, providing the industry with different qualities to optimize the usage of the resource depending on the final product. With respect to the percentage of rejection, only Scots pine presented worse values in machine grading (for portable equipment) than visual grading.

**Author Contributions:** Conceptualization, V.B. and G.M.; methodology, G.M. and G.C.; validation, G.Í.-G. and V.B.; formal analysis, G.M.; investigation, G.M., G.C. and V.B.; data curation, G.M.; writing—original draft preparation, G.M.; writing—review and editing, V.B. and G.Í.-G.; supervision, V.B.; project administration, V.B. All authors have read and agreed to the published version of the manuscript.

**Funding:** This research was funded by: (1) the Institute for Business Competitiveness of Castilla y León, co-financed by FEDER funds (CCTT2/18/SO/0001: Research for the modernization of manufacturing processes and timber grading to obtain technological products); (2) Ministerio de Agricultura, Pesca, y Alimentación through the program AEI-AGRI (20180020012467: SIGCA. Forest management systems in forests producing quality wood); (3) Interreg POCTEFA 2014-2020 program co-financed by FEDER funds through Coopwood project; and (4) Interreg SUDOE program co-financed by FEDER funds through Eguralt project (SOE4/P1/E1115: Application and dissemination of innovative solutions for the promotion of mid-rise timber construction in the SUDOE area).

**Data Availability Statement:** Not applicable.

**Acknowledgments:** The authors would like to thank José Luis Villanueva, responsible of the wood industry group at Cesefor, for the Sigca and Coopwood project administration, and to the laboratory

staff of Cesefor, Laura Gómez, Luis Molina, Cristian Ribas, and Álvaro Sánchez, for the collaboration in testing.

**Conflicts of Interest:** The authors declare no conflict of interest.

## Abbreviations

| Symbol | Description |
| --- | --- |
| N | Number of specimens |
| $E_{0,m}$ | Modulus of elasticity parallel to grain |
| $f_m$ | Bending strength |
| $\rho$ | Density |
| IP | Indicative property |
| L | Specimen length |
| $f_0$ | Frequency of first longitudinal vibration mode |
| CF | Correction factor of wave speed based on moisture content |
| $\rho_{12}$ | Density corrected to a moisture content of 12% |
| u | Moisture content |
| $E_{0,mean,EN338}$ | Modulus of elasticity given by EN338 for a specific strength class |
| $f_k$ | Characteristic strength of a sample |
| $k_v$ | Factor considering the variability between subsamples, given by EN384 |
| $f_{05}$ | 5th percentile of the strength |
| $f_{m,k}$ | Characteristic bending strength of a sample |
| $\overline{E_0}$ | Average modulus of elasticity of a sample |
| $\rho_{mean}$ | Average density of a sample |
| $\rho_k$ | Characteristic density of a sample |

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
