# Peer review of "Evaluation of Yield Improvements in Machine vs. Visual Strength Grading for Softwood Species"

_forests, doi:10.3390/f13122021_

Round 1
Reviewer 1 Report
It is interesting to see a study that covers both visual and machine grading on the same dataset, but the introduction doesn’t really tell the reader why such a comparison is interesting, or why it was done.
Comparison of yields is, I suppose, useful for a reader who might be thinking about grading Spanish pine, but for a more general audience it would perhaps be more useful also to look a little into how differently the visual and machine grading sort the pieces. Do they tend to agree in which pieces are assigned to which grade or not? Would we expect them to agree? Do they need to agree for grading to work?
Abstract. Lines 14-15
It should be made clear here that the assignment to strength classes described is for Spanish grown timber. For clarity it should say “European redwood” instead of just redwood.
Introduction. Line 30
The word “previously” should be deleted because it is not necessary (and a little confusing)
Line 35
It is confusing to mention machine grading here. Better to say that visual grading can involve measurement of density.
Line 47
I think it is too much to claim that visual grading is not reliable. It may be considered safe, but it might not result in the desired yield and production speed.
Line 67
There is a missing space in “mtgBATCH 966 from”
Line 85
Perhaps it is necessary here to say that one reason the portable grading machines (I assume you mean portable and not just the handheld type) is that it does not benefit from a bonus to the strength (kv - that you describe later) that the inline grading machines do. “Production speed” is perhaps clearer than “timing” if that is what you mean.
Sampling, general
I think you should add that the sampling was representative of the timber to be graded in production.
You don’t say directly that P. pinaster ssp. Mesogensis is not included in your sampling or covered by the grading settings. I presume that is what you mean to communicate.
Section 2.2 Line 127
I think you should say that the handheld version of the machine gives the same frequency and mass measurement as the inline version. Not every reader would know that.
Section 2.3 Line 134
You should reference EN384
Line 137 the local modulus of elasticity needs a little more explanation for readers not familiar with EN408.
Section 2.4 Line 159
You should explain that density is also adjusted for moisture content.
Section 2.5 general
The text here is a bit confusing. In particular you mention the 0.95 factor on stiffness twice. You should also explain to the reader that you are describing what the standards require for the calculation.
It would be helpful to have a list of what the symbols mean
Section 3.1, line 234
It is confusing to say that a strength class can be assigned without visual or machine grading. It is clearer to say that, before grading, the timber has the required properties for those strength classes.
Section 3.2 general
You don’t mention EN384 here but I assume you mean that these values come from EN384 and how it tells you to calculate with values obtained from EN14358
Section 3.5 line 318
Radiata is misspelled
Author Response
It is interesting to see a study that covers both visual and machine grading on the same dataset, but the introduction doesn’t really tell the reader why such a comparison is interesting, or why it was done.
Comparison of yields is, I suppose, useful for a reader who might be thinking about grading Spanish pine, but for a more general audience it would perhaps be more useful also to look a little into how differently the visual and machine grading sort the pieces. Do they tend to agree in which pieces are assigned to which grade or not? Would we expect them to agree? Do they need to agree for grading to work?
Since the mechanical grading with homologated machines according to CEN is relatively new and the results are confidential for the committee, there is a need to quantify how the machine grading improves the strength grading yield with respect to visual grading. It was clarified in the introduction and title. The agreement of the grading between both methods suggested by the reviewer is a topic of interest, which probably ca be discussed in future works, but currently, it is out of the paper’s reach. It should be noticed that the results do not need to agree for the grading to work and that it does not compromise the discussion of the obtained results.
Abstract. Lines 14-15
It should be made clear here that the assignment to strength classes described is for Spanish grown timber. For clarity it should say “European redwood” instead of just redwood.
Thanks for the comment. European redwood has been included in the manuscript as Scots pine according to EN 13556:2003
Introduction. Line 30
The word “previously” should be deleted because it is not necessary (and a little confusing)
Changed
Line 35
It is confusing to mention machine grading here. Better to say that visual grading can involve measurement of density.
Ok
Line 47
I think it is too much to claim that visual grading is not reliable. It may be considered safe, but it might not result in the desired yield and production speed.
It was added that it is terms of productivity
Line 67
There is a missing space in “mtgBATCH 966 from”
Ok, corrected
Line 85
Perhaps it is necessary here to say that one reason the portable grading machines (I assume you mean portable and not just the handheld type) is that it does not benefit from a bonus to the strength (kv - that you describe later) that the inline grading machines do. “Production speed” is perhaps clearer than “timing” if that is what you mean.
Thanks for the suggestion, which improves the understanding of the manuscript. “Handheld” was changed to “portable”, “timing” was changed to “grading speed”, and the “cost” associated with “personnel hours”.
Sampling, general
I think you should add that the sampling was representative of the timber to be graded in production.
You don’t say directly that P. pinaster ssp. Mesogensis is not included in your sampling or covered by the grading settings. I presume that is what you mean to communicate.
The text was modified to clarify it.
Section 2.2 Line 127
I think you should say that the handheld version of the machine gives the same frequency and mass measurement as the inline version. Not every reader would know that.
We agree with your comment. A clarification was added in the manuscript.
Section 2.3 Line 134
You should reference EN384
The reference was added
Line 137 the local modulus of elasticity needs a little more explanation for readers not familiar with EN408.
A brief comment was added
Section 2.4 Line 159
You should explain that density is also adjusted for moisture content.
The explanation was added
Section 2.5 general
The text here is a bit confusing. In particular you mention the 0.95 factor on stiffness twice. You should also explain to the reader that you are describing what the standards require for the calculation.
It would be helpful to have a list of what the symbols mean
A clarification was made on the manuscript for the 0.95 factor, and the list of symbols was added at the start of the paper
Section 3.1, line 234
It is confusing to say that a strength class can be assigned without visual or machine grading. It is clearer to say that, before grading, the timber has the required properties for those strength classes.
We agree with your comment. It was changed to reflect that the values are equivalent to the strength class
Section 3.2 general
You don’t mention EN384 here but I assume you mean that these values come from EN384 and how it tells you to calculate with values obtained from EN14358
Yes, it was clarified.
Section 3.5 line 318
Radiata is misspelled
Ok
Reviewer 2 Report
This is an excellent manuscript, which summaries a very useful study. My only suggestion would be to include a recent publication on machine grading of lumber:
Entsminger, E.D.; Brashaw, B.K.; Seale, R. Daniel; Ross, R.J. 2020. Machine grading of lumber--practical concerns for lumber producers. General Technical Report FPL-GTR-279. Madison, WI: U.S. Department of Agriculture, Forest Service, Forest Products Laboratory. 66p.
Machine grading of lumber—practical concerns for lumber producers (usda.gov)
It is an excellent document on machine grading.
This publication is the latest (third) edition of this publication, the original (first edition) published in 1977 (Machine Stress Rating---Practical Concerns for Lumber Producers by William Galligan, Delos V. Snodgrass and Gerald W. Crow. Please note that I cannot find an electronic version of this edition.
The second edition (2000), Machine Grading of Lumber--Practical Concerns For Lumber Producers (usda.gov), was authored by William Galligan and Kent McDonald.
I only suggest it be included in the introductory material (I believe in the third paragraph) because it would provide many readers not knowledgeable in the area a reference document that is a comprehensive review of machine grading and a solid scientific/technical review/foundation.
Note that is a document that is updated, hence the third edition---authorship is based, in-part, on who is on staff at the time.
Author Response
Thank you for the suggestion. It was a really interesting read, we added it in the introduction.
Reviewer 3 Report
Paper mainly reports results from laboratory test activity which is carried out in accordance with European standards and it is usually aimed to certify, by competent Technical Groups, machines for timber strength grading. Therefore, the content is overall correct but it is not innovative from a scientific point of view.
It is not clear what is the aim of the work and what the authors want to add to the already established knowledge on the topic.
A suggestion to the authors could be to use the data collected during laboratory tests to better described the resource, to analyze the correlations between parameters detected by grading machines and the outputs. Moreover, the results of the visual classification could be better investigated in order to highlight if improvements could be introduced with respect to the classification rules currently included in the national standard.
Author Response
Thank you for your comments, we made changes to point out the objectives of the research paper considering also the comments of the other reviewers.
Since the mechanical grading with homologated machines according to CEN is relatively new and the results are confidential for the committee, there is a need to quantify how the machine grading improves the strength grading yield with respect to visual grading and make this information available to the academic and industrial sector. It was clarified in the introduction and title. Your proposal for the visual grading study is of interest and will be considered for future publications, nevertheless, quantifying how machine grading improves the yields vs. visual grading is a current need. It is the reason why this paper was submitted to this special issue of the Forests journal focused on timber characterization.
Round 2
Reviewer 3 Report
Dear Authors,
considering your response, I do suggest to see the following paper dealing with the same topic to have a broader literature analysis and also in order to compare the your results to what was previously observed:
Brunetti et al. 2016. Visual and machine grading of larch (Lartix decidua Mill.) structural timber from the Italiaqn Alps. https://doi.org/10.1617/s11527-015-0676-5